# Public awareness of and responses to media coverage of invitation errors in the Breast Screening Programme in England: a cross-sectional population survey

Alex Ghanouni, Christian von Wagner, Jo Waller

Research Department of Behavioural Science and Health, University College London, London, UK

**Correspondence to**
Dr Jo Waller; j.waller@ucl.ac.uk

## ABSTRACT

**Objectives** In May 2018, the British Health Secretary announced the 'serious failure' that 450 000 women had missed out on invitations to breast screening in England, leading to extensive media coverage. This study measured public awareness of the story and tested for associated factors (eg, educational level and trust in the National Health Service (NHS)).

**Design** A computer-assisted face-to-face survey in June 2018.

**Setting** Participants completed the survey in their homes.

**Participants** Males and females aged 16 years or older in England.

**Primary and secondary outcome measures** Awareness of aspects of the media coverage and reported statistics. Other data included demographics (eg, ethnicity), awareness of unrelated contemporaneous news stories, trust in participants' general practitioners (GPs) and the NHS, and (among women) worry about breast cancer and future breast screening intentions.

**Results** Descriptive statistics showed that 67% of 1894 participants reported being aware of the media coverage. Regression analyses showed that those who were aware of other news stories, were white British and had a higher level of education or social grade were more likely to be aware. In contrast, only 36% correctly identified at least one of two headline statistics. This study did not find evidence that awareness was negatively associated with trust in participants' GPs or the NHS, breast cancer worry or future breast cancer screening intentions.

**Conclusions** Awareness of the breast screening news story was high but recall of reported statistics was much lower: the public may have retained only the gist of quantitative information. Associations between story awareness and attitudes or behaviour were not apparent.

## INTRODUCTION

On 2 May 2018, the Health Secretary in Great Britain, Jeremy Hunt, made an unanticipated statement to the House of Commons regarding '*a serious failure…in the national Breast Screening Programme.*' Mr Hunt stated that since 2009, '*a computer algorithm failure*'

### Strengths and limitations of this study

► This study builds on previous research on media coverage around public health concerns by measuring levels of awareness among the general public and testing for characteristics associated with awareness.

► The survey was carried out shortly after media coverage of the announcement began, when awareness and knowledge were likely to be at their highest.

► Associations between awareness of media coverage and, for example, greater worry about breast cancer and lower trust in the National Health Service were not apparent but type II error cannot be excluded.

► Tests for associations between awareness of media coverage and screening behaviour were based on intended future uptake; actual uptake may differ.

had resulted in approximately 450 000 women not being invited to their final regular breast screening appointment (ie, when they were aged 68–71 years). He indicated to the House that '*[the] current best estimate based on statistical modelling…is that there may be between 135 and 270 women who had their lives shortened as a result*' and that women affected '*will automatically be sent an invitation to a catch-up screening.*'[1] News of this statement was reported extensively in the national media (eg, refs 2–4) and prompted a volume of follow-up commentary from academics aiming to add context to this story. For example, some raised arguments that breast screening has no effect on all-cause mortality and risks resulting in overdiagnosis.[5 6]

Awareness of health-related media coverage is likely to be very high among academics and clinicians who are professionally invested in the topic. However, research is lacking on the prevalence of awareness of this type of news among the general public. In the absence of

empirical data, it might be hypothesised to be either high (eg, because mainstream media coverage has an extremely wide reach) or generally low (because members of the public are more focused on their personal priorities or do not have a specific interest in health news).

The level of public awareness of health media coverage is significant because it represents the proportion of people who may be influenced by it: previous research has found that media coverage of cancer-related stories in the UK has appreciable public health implications. For example, there is evidence that the cervical cancer diagnosis and death of a young female celebrity, Jade Goody, influenced women's cervical cancer screening decisions and temporarily increased uptake and diagnoses of high-grade cervical neoplasia.[7–10] Similarly, uptake of the colorectal screening programme increased following coverage of the UK Flexible Sigmoidoscopy Screening Trial.[11 12] This coverage often contained elements likely to be perceived highly favourably by the general public such as the fact that it was a 5 min, one-off test that could save thousands of lives. In addition, the word 'breakthrough' was often featured.[13–16] Comparable findings have been reported by studies of preplanned media messages such as Public Health England's 'Be Clear on Cancer' campaigns, which aim to increase cancer symptom awareness. These were associated with an increase in symptomatic attendance at General Practices and referrals to secondary care.[17–19]

In these cases, media coverage was associated with an increase in healthcare usage. However, news about an error in the screening programme may have had adverse effects, such as reducing trust in the National Health Service (NHS; with corresponding negative implications for help seeking), more frequent worry about breast cancer and being less inclined to have breast screening in future. To our knowledge, this possibility has not been investigated by research to date.

This study surveyed awareness of the coverage shortly after the announcement (when conscious recall was likely to be highest) in a large, sociodemographically diverse sample of the general public. In order to make a more complete assessment of this awareness, we also measured knowledge of the relevant statistics most commonly reported as part of the story (ie, the number of women estimated to have missed an invitation and to have had their lives shortened) since these were a key factor in making a personal assessment of the scale and severity of the invitation errors. We also recognised that people's concerns about the initial coverage may have been moderated by follow-up commentary noting issues around overdiagnosis and all-cause mortality in breast screening. We used these measures to conduct an exploratory analysis of variables associated with awareness of the media coverage, including education, gender and awareness of other news stories that were reported around the same time. We also tested the hypotheses that awareness of the breast screening media coverage would be associated with lower trust in participants' general practitioners (GPs) and the NHS and (in women) more frequent worry about

breast cancer and being less likely to intend to participate in breast screening in future.

## METHODS

### Design

A market research agency (Kantar TNS UK) collected data in two waves of sampling between 6 and 10 June 2018 (ie, less than 6 weeks after the initial news story). The survey questions formed one module within a weekly face-to-face computer-assisted omnibus survey on a wider range of topics. Random location sampling was used to identify target households based on the 2011 Census and Postcode Address File. At each location, quotas were set with the aim of achieving national representativeness based on working status, children in the household, gender and age.

The full survey is included in online supplementary appendix 1. Participants were initially shown a computer screen with text introducing the study and asking for their consent to participate. They were also given an information card containing debrief text and directions to further information about breast screening.

### Participants

Eligible participants were all males and females in England aged 16 years or older who consented to take part in this module of the survey. The sample includes women eligible for breast screening (ie, aged 47–73 years) and also members of the general population (males and females aged 16 years or older) since it was hypothesised that awareness of the story had the potential to negatively affect perceptions of other health services, irrespective of whether participants were affected directly. Sample size was based on budgetary constraints and the number of participants who could be approached no more than 6 weeks after the initial news story.

### Patient and public involvement

Since the results of the study were expected to be highly time sensitive, rapid data collection was prioritised over involving patients and the public in the design and conduct of the study. In order to minimise data protection issues, survey responses were received by the research team in anonymised format, meaning that it is not possible to disseminate study results to participants.

### Measures

#### Demographics

General background information included participants' self-reported age (in years), gender, ethnic origin, marital status, education, social grade,[20] employment status and urban or rural area type.

#### Cancer and breast screening experience, and attitudes towards screening

Participants were asked whether they had been diagnosed with any of several types of cancer themselves. Women aged 47 years or older were also asked if they had ever

been (1) invited to and (2) participated in the Breast Screening Programme.

Participants were asked about their attitudes towards screening via a previously used question,[21] '*Routine screening means testing healthy people to find cancer before they have any symptoms. Do you think routine cancer screening tests for healthy people are almost always a good idea?*' Response options were '*yes*', '*no*' and '*not sure*'.

### Awareness of the breast screening news story

Participants were asked to read a brief summary of the story (see online supplementary appendix 1, Q7), the main details of which were derived from the primary story on the topic on the BBC news website.[4] This was followed by the question, '*Do you recall seeing or hearing anything about this news story before now?*' Response options were '*yes*', '*no*' and '*not sure*'.

It was anticipated that directions of associations with awareness may depend on the specific parts of the story of which participants were aware. Consequently, participants who reported being aware of the main news story were also asked about their awareness of issues relating to all-cause mortality and overdiagnosis using two further summaries (see online supplementary appendix 1, Q14 and Q15), derived from two sources.[5 6]

Questions for assessing awareness were the same as previous. Participants reporting awareness of the news story were also asked where they saw or heard it and whether they discussed or shared it with anyone else. They were also asked two questions on the key statistics reported based on the following summaries:

The Health Secretary, Jeremy Hunt, gave an estimate of the number of women who had failed to get invitations since 2009.

The Health Secretary also gave an estimate, based on computer modelling, of the number of women who may have had their lives shortened.

For both, the question was '*Which of the following do you think is the estimate that he gave?*' For the first question, response options consisted of the true estimate (450 000) and three alternatives that were orders of magnitude higher or lower (4500, 45 000 and 4 500 000). Similarly, response options for the second question consisted of the correct answer (between 135 and 270) and alternatives that were either an order of magnitude higher (1350 and 2700), lower (13 and 27), or both higher and lower (13 and 2700). Response order was presented in one of two different ways for each participant (determined at random) to reduce potential order effects.

### Awareness of news stories unrelated to breast screening

Awareness of other news stories was measured by asking participants to read two further summaries (one on a volcano eruption in Hawaii; one on local council elections in England; see online supplementary appendix 1, Q19 and Q20). This was followed by the same measure of awareness as in previous questions. Main details

were derived from the primary stories on the BBC news website.[22 23] These two stories were selected for comparison because they were reported around the same time and also consisted of specific, definable events.

### Trust in health services

Participants were asked two questions based on previously used items,[24 25] '*In general, how much do you trust…*' (1) '*…your general practitioner?*' and (2) '*…the NHS?*' Response options for both were '*not at all*', '*a little*', '*somewhat*', '*a lot*' and '*not sure*'.

### Frequency of breast cancer worry

Breast cancer worry (among women) was measured using an item based on one previously used,[26] '*How often do you worry about your chances of getting breast cancer yourself?*' Response options were '*never*', '*occasionally*', '*sometimes*', '*often*', '*very often*', '*not sure*' and '*prefer not to say*'.

### Breast screening intentions

Women aged 16–69 years were asked, '*Do you think you will go for breast screening when you are next offered it?*' Response options were '*yes, definitely*', '*yes, probably*', '*no, probably not*' and '*no, definitely not*'.

### Analysis

Participant characteristics and awareness about the news stories are reported using descriptive statistics. Responses of '*prefer not to say*' were excluded, as were responses of '*not sure*' for ordinal variables. Other responses of '*not sure*' were grouped with '*no*'. Ethnicity was dichotomised into '*white British*' and '*other groups*'; social grades were grouped into '*A or B*', '*C1*', '*C2*' and '*D or E*'. For education, '*trade apprenticeships*' were grouped with '*other qualifications*'. Responses to measures of invitations to and participation in breast screening were coded into '*not eligible or not invited*', '*invited, never taken part*' and '*taken part*'.

One exploratory regression model tested for variables potentially associated with whether people responded to the survey. Three exploratory regression models tested for variables potentially associated with (1) awareness of the breast screening news; and stating correctly the number of women who were (2) not invited for screening and (3) estimated to have had their lives shortened. A further four regression models tested the null hypotheses that awareness of the breast screening news story was not associated with trust in (4) participants' GPs and (5) the NHS in the whole sample; and (6) frequency of worry about breast cancer and (7) intentions to participate in breast screening in future among women aged 70 years or less, after adjusting for covariates.

For the model assessing variables associated with responding to the questionnaire, the main variables of interest were recruitment wave, gender, ethnicity, marital status, social grade, employment status, area type and age (since these were the variables where data were available for both participants and non-participants). For the four main exploratory models and hypothesis testing models, independent variables were as above with the addition of

other available measures (listed in tables) where multicollinearity was not an appreciable issue (ie, variance inflation factors <10). Age was included in models as either a continuous variable or divided into age groups (where a Box-Tidwell procedure found evidence that the assumption of linearity was not met; p<0.05). Frequency of worry about breast cancer was also included in the model of future breast screening intentions.

For models testing hypotheses, responses on measures of awareness of the breast screening story were coded into a single nominal variable with five levels: (1) '*unaware of the story*', (2) '*aware of the main story only*', (3) '*aware of the main story and all-cause mortality follow-up commentary*',(4) '*aware of the main story and overdiagnosis follow-up commentary*', (5) '*aware of the main story and both follow-up commentaries*'.

Ordinal logistic regression was attempted in the first instance where dependent variables were ordinal. Tests of parallel lines suggested that the assumption of proportional odds was generally not met (p<0.0005) and there were few cases in some cells. Hence, dependent variables were dichotomised and binary logistic regression was used. Participants with missing data on variables of interest were not included in models.

## RESULTS
### Participant characteristics
A total of 2681 participants began the survey module. 787 (29.4%) opted out, leaving 1894 participants who provided data. Mean age was 50.8 years (SD: 20.5). Characteristics are described in online supplementary appendix 2 table A. Response to the survey module questions was associated with all variables in the model, except for area type (online supplementary appendix 2 table B). Participants approached for the omnibus survey were more likely to respond to this survey module if they were invited in wave 1 (vs wave 2), female (vs male), white British (vs other groups), married, living as a couple, or widowed, divorced or separated (vs single), in higher social grades (vs grade D or E), working (vs not working) and younger.

### Awareness of news stories, sources of information and variables associated with awareness of the breast screening media coverage
There were 1264 out of 1894 participants (66.7%) who reported being aware of the main news story (online supplementary appendix 2 table A) and relatively few reported being aware of follow-up commentaries: 438/1264 (34.7%) and 367/1264 (29.0%) recognised the commentaries on all-cause mortality and overdiagnosis, respectively. Two hundred and fifty out of 1264 (19.8%) were aware of both. Nine hundred and seventy-one out of 1264 (76.8%) and 271/1264 (21.4%) encountered the story on television and radio, respectively (participants could select more than one). One hundred and sixty-nine out of 1264 (13.4%) and 134/1264 (10.6%) encountered the story in print newspapers and online news websites (online supplementary appendix 2 table C). Other news

sources were used relatively rarely, for example, 68/1264 (5.4%) heard the story from social media websites. Four hundred and fifty out of 1264 (35.6%) reported discussing or sharing the story with someone else.

Participants were more likely to be aware of the story if they were aware of either of the other two news stories. Awareness of the three stories was highly inter-related: 824/1894 participants (43.5%) were aware of all three news stories and a further 196/1894 (10.3%) reported not being aware of any. Only 323/1894 (17.1%) were aware of just one of the three stories and only 106/1894 participants (5.6%) were aware of the news about breast screening, specifically. Participants were also more likely to be aware of the breast screening news story if they were white British, older, had higher levels of education or social grade. Participants were less likely to be aware if they believed that screening was almost always a good idea. All other p values were ≥0.207 (table 1).

### Awareness of statistics from the breast screening media coverage and variables associated with awareness among participants who reported being aware of the story
Only 233 (18.4%) of the 1264 participants who reported being aware of the story correctly recognised the number of women who had not been invited and only 268 (21.2%) correctly recognised the estimated number of women who had their lives shortened. Eight hundred and nine (64.0%) did not correctly identify either statistic and only 3.6% correctly identified both (table 2). The model testing for demographic and psychological variables associated with correctly identifying either set of statistics found only weak evidence against the null hypothesis for all characteristics (p values were ≥0.087 and ≥0.062 in the respective models; data not shown).

### Awareness of media coverage and participants' trust in their GPs and the NHS
In both these models, there was only weak evidence against the null hypothesis. Table 3 shows the main results of binary logistic regression models consisting of 1746 participants (p=0.729 and p=0.290). Full results of the model are presented in online supplementary appendix 2 tables D and E.

### Awareness of media coverage and frequency of worry about breast cancer
Table 4 shows that there was only weak evidence against the null hypothesis (n=700; p=0.198). Full results are included in online supplementary appendix 2 table F.

### Awareness of media coverage and future breast screening intentions
Table 5 shows that there was only weak evidence against the null hypothesis for this analysis (n=700; p=0.108). Full results are included in online supplementary appendix 2 table G. Numbers of participants with missing data for each variable are shown in online supplementary appendix 2 table H.

**Table 1** Full results of the binary logistic regression model testing for variables associated with awareness of the breast screening news story

| Characteristic | Total (n=1792) | Aware versus not aware of the breast screening story (or not sure): n (%) | | Aware of the screening story (vs not aware or not sure) | |
|---|---|---|---|---|---|
| | | Not aware/sure (n=587; 32.8%) | Aware (n=1205; 67.2%) | Adjusted OR 95% CI | P value |
| **Recruitment wave** | | | | | |
| Wave 2: 20–26 June | 570 | 185 (32.5) | 385 (67.5) | 1.02 0.79 to 1.31 | 0.907 |
| vs Wave 1: 6–10 June | 1222 | 402 (32.9) | 820 (67.1) | | |
| **Age** | | | | Overall: **<0.0005** | |
| 65+ | 549 | 111 (20.2) | 438 (79.8) | 7.77 4.52 to 13.38 | **<0.0005** |
| 55–64 | 252 | 53 (21.0) | 199 (79.0) | 6.75 3.92 to 11.63 | **<0.0005** |
| 45–54 | 241 | 47 (19.5) | 194 (80.5) | 7.70 4.56 to 13.00 | **<0.0005** |
| 35–44 | 248 | 88 (35.5) | 160 (64.5) | 3.60 2.22 to 5.84 | **<0.0005** |
| 25–34 | 275 | 142 (51.6) | 133 (48.4) | 2.00 1.27 to 3.14 | **0.003** |
| vs 16–24 | 227 | 146 (64.3) | 81 (35.7) | | |
| **Gender** | | | | | |
| Male | 771 | 234 (30.4) | 537 (69.6) | 1.00 0.74 to 1.35 | 0.999 |
| vs Female | 1021 | 353 (34.6) | 668 (65.4) | | |
| **Ethnicity** | | | | | |
| White British | 1491 | 415 (27.8) | 1076 (72.2) | 3.00 2.20 to 4.09 | **<0.0005** |
| vs Other groups | 301 | 172 (57.1) | 129 (42.9) | | |
| **Marital status** | | | | Overall: 0.914 | |
| Married/living as a couple | 985 | 279 (28.3) | 706 (71.7) | 1.07 0.78 to 1.47 | 0.672 |
| Widowed/divorced/separated | 354 | 84 (23.7) | 270 (76.3) | 1.06 0.70 to 1.60 | 0.792 |
| vs Single | 453 | 224 (49.4) | 229 (50.6) | | |
| **Highest level of education** | | | | Overall: **0.001** | |
| Graduate level/above | 501 | 131 (26.1) | 370 (73.9) | 2.08 1.34 to 3.23 | **0.001** |
| A levels/AS levels/equivalents | 448 | 162 (36.2) | 286 (63.8) | 1.80 1.19 to 2.73 | **0.006** |
| GCSEs/equivalents | 440 | 156 (35.5) | 284 (64.5) | 1.36 0.92 to 2.00 | 0.120 |
| Trade apprenticeships/other | 89 | 39 (43.8) | 50 (56.2) | 0.75 0.42 to 1.32 | 0.316 |
| vs No formal qualifications | 314 | 99 (31.5) | 215 (68.5) | | |
| **Social grade** | | | | Overall: **<0.0005** | |
| Grade A or B | 326 | 53 (16.3) | 273 (83.7) | 2.44 1.59 to 3.73 | **<0.0005** |
| Grade C1 | 511 | 165 (32.3) | 346 (67.7) | 1.41 1.02 to 1.95 | **0.037** |
| Grade C2 | 394 | 142 (36.0) | 252 (64.0) | 1.13 0.81 to 1.58 | 0.469 |
| vs Grade D or E | 561 | 227 (40.5) | 334 (59.5) | | |
| **Employment status** | | | | | |

**Table 1** Continued

| Characteristic | Total (n=1792) | Aware versus not aware of the breast screening story (or not sure): n (%) | | Aware of the screening story (vs not aware or not sure) | |
|---|---|---|---|---|---|
| | | Not aware/sure (n=587; 32.8%) | Aware (n=1205; 67.2%) | Adjusted OR 95% CI | P value |
| Working | 823 | 287 (34.9) | 536 (65.1) | 0.91 0.68 to 1.22 | 0.909 |
| vs Not working | 969 | 300 (31.0) | 669 (69.0) | | |
| Area type | | | | | |
| Urban | 1458 | 476 (32.6) | 982 (67.4) | 1.21 0.90 to 1.64 | 0.207 |
| vs Rural | 334 | 111 (33.2) | 223 (66.8) | | |
| Personal diagnosis of cancer | | | | | |
| Yes | 150 | 34 (22.7) | 116 (77.3) | 1.18 0.74 to 1.86 | 0.490 |
| vs No | 1642 | 553 (33.7) | 1089 (66.3) | | |
| Personal experience of breast screening | | | | Overall: 0.552 | |
| Taken part | 425 | 90 (21.2) | 335 (78.8) | 0.92 0.60 to 1.41 | 0.705 |
| Invited, never taken part | 55 | 13 (23.6) | 42 (76.4) | 0.66 0.32 to 1.39 | 0.276 |
| vs Not eligible or not invited | 1312 | 484 (36.9) | 828 (63.1) | | |
| Belief that screening is almost always a good idea | | | | | |
| Yes | 1649 | 547 (33.2) | 1102 (66.8) | 0.59 0.38 to 0.94 | **0.025** |
| vs No or not sure | 143 | 40 (28.0) | 103 (72.0) | | |
| Awareness of volcano news | | | | | |
| Yes | 1367 | 325 (23.8) | 1042 (76.2) | 3.14 2.39 to 4.12 | **<0.0005** |
| vs No or not sure | 425 | 262 (61.6) | 163 (38.4) | | |
| Awareness of election news | | | | | |
| Yes | 1138 | 292 (25.7) | 846 (74.3) | 1.37 1.06 to 1.75 | **0.014** |
| vs No or not sure | 654 | 295 (45.1) | 359 (54.9) | | |
| General level of trust in the NHS | | | | Overall: 0.485 | |
| A lot | 969 | 308 (31.8) | 661 (68.2) | 0.59 0.29 to 1.18 | 0.132 |
| Somewhat | 599 | 193 (32.2) | 406 (67.8) | 0.63 0.31 to 1.27 | 0.196 |
| A little | 169 | 69 (40.8) | 100 (59.2) | 0.58 0.27 to 1.25 | 0.166 |
| vs Not at all | 55 | 17 (30.9) | 38 (69.1) | | |

Bold values denotes p<.05
A level, Advanced level; AS level, Advanced Subsidiary level; GCSE, General Certificate of Secondary Education; NHS, National Health Service.

## DISCUSSION

Previous studies have found evidence that media messages can increase usage of a range of healthcare services (eg, refs 7–10 12 17–19). Awareness of this story about errors in the Breast Screening Programme was hypothesised to have the potential for a range of negative effects. However, the results of this study did not provide strong evidence against the null hypothesis for any associations tested. To the extent that these results reflect an absence of harms, this is reassuring: we did not find evidence that awareness of the story reduced trust in the NHS or participants' GPs, increased frequency of worry about breast cancer, or negatively affected future breast screening intentions. If this is the case, it may be partly attributable to the

**Table 2** Descriptive statistics of participants' responses about key statistics in the breast screening media coverage; correct responses were '450 000' and '135–270'

| | n (% of total; 95% CI) (n=1264) | | | | | |
|---|---|---|---|---|---|---|
| Number of women who did not receive their final invitation… | Number of women who may have had their life shortened. Between… | | | | | |
| | 135–270 | 13–27 | 13–2700 | 1350–2700 | Not sure | Total |
| 450 000 | 46 (3.6) | 6 (0.5) | 79 (6.3) | 71 (5.6) | 31 (2.5) | 233 (18.4) |
| 4500 | 68 (5.4) | 20 (1.6) | 28 (2.2) | 22 (1.7) | 30 (2.4) | 168 (13.3) |
| 45 000 | 130 (10.3) | 22 (1.7) | 76 (6.0) | 86 (6.8) | 54 (4.3) | 368 (29.1) |
| 4 500 000 | 3 (0.2) | 1 (0.1) | 10 (0.8) | 20 (1.6) | 4 (0.3) | 38 (3.0) |
| Not sure | 21 (2.1) | 5 (0.4) | 15 (1.2) | 12 (0.9) | 404 (32.0) | 457 (36.2) |
| Total | 268 (21.2) | 54 (4.3) | 208 (16.5) | 211 (16.7) | 523 (41.4) | |

news story saying little to reduce the perceived benefits of breast screening itself, in contrast to media coverage of, for example, the independent review of breast cancer screening, which reported on the issue of overdiagnosis extensively.[27 28] Relatedly, the present study found that awareness was notably lower for follow-up commentaries on the shortcomings of breast screening, compared with the main story. In addition, the framing of the story may have been expected to reinforce the perceived benefits of screening by indicating that missing screening had

**Table 3** Testing for an association between awareness of the breast screening media coverage and trust in (1) participants' GPs and (2) the NHS*

| General level of trust in participants' GPs | | A lot versus not at all; a little; somewhat: n (%) | | A lot (vs less than a lot) | |
|---|---|---|---|---|---|
| Characteristic | Total (n=1746) | Less than a lot (n=781; 44.7%) | A lot (n=965; 55.3%) | Adjusted OR 95% CI | P value |
| Screening story awareness | | | | Overall: 0.729 | |
| Aware of the main story and both follow-up commentaries | 238 | 98 (41.2) | 140 (58.8) | 1.10 (0.74 to 1.64) | 0.653 |
| Aware of the main story and overdiagnosis follow-up | 172 | 66 (38.4) | 106 (61.6) | 1.31 (0.85 to 2.03) | 0.218 |
| Aware of the main story and all-cause mortality follow-up | 107 | 49 (45.8) | 58 (54.2) | 1.21 (0.73 to 2.02) | 0.459 |
| Aware of the main story only | 655 | 280 (42.7) | 375 (57.3) | 1.17 (0.88 to 1.57) | 0.283 |
| vs Unaware of the story | 574 | 288 (50.2) | 286 (49.8) | | |

| General level of trust in the NHS | | A lot versus not at all; a little; somewhat: n (%) | | A lot (vs less than a lot) | |
|---|---|---|---|---|---|
| Characteristic | Total (n=1746) | Less than a lot (n=803; 46.0%) | A lot (n=943; 54.0%) | Adjusted OR 95% CI | P value |
| Screening story awareness | | | | Overall: 0.290 | |
| Aware of the main story and both follow-up commentaries | 238 | 102 (42.9) | 136 (57.1) | 0.87 (0.59 to 1.30) | 0.503 |
| Aware of the main story and overdiagnosis follow-up | 172 | 76 (44.2) | 96 (55.8) | 0.78 (0.51 to 1.21) | 0.267 |
| Aware of the main story and all-cause mortality follow-up | 107 | 57 (53.3) | 50 (46.7) | 0.58 (0.35 to 0.97) | 0.039 |
| Aware of the main story only | 655 | 299 (45.6) | 356 (54.4) | 0.81 (0.60 to 1.09) | 0.160 |
| vs Unaware of the story | 574 | 269 (46.9) | 305 (53.1) | | |

*Results are adjusted based on the following covariates: recruitment wave, age (age group in the model of trust in the NHS), gender, ethnicity, marital status, highest level of education, social grade, employment status, area type, personal diagnosis of cancer, personal experience of breast screening, belief that screening is almost always a good idea, awareness of volcano news, awareness of election news, general level of trust in the NHS (general level of trust in participants' GPs in the model of trust in the NHS). Full results of the model are reported in the online supplementary appendix 2.
GP, general practitioner; NHS, National Health Service.

**Table 4** Testing for an association between awareness of the breast screening media coverage and frequency of worry about breast cancer*

| Characteristic | Total (n=700) | Never; occasionally versus sometimes; often; very often: n (%) | | Sometimes; often; very often (vs never; occasionally) | |
| | | Never; occasionally (n=441; 63.0%) | Sometimes; often; very often (n=259; 37.0%) | Adjusted OR 95% CI | P value |
|---|---|---|---|---|---|
| Screening story awareness | | | | Overall: 0.198 | |
| Aware of the main story and both follow-up commentaries | 88 | 65 (73.9) | 23 (26.1) | 0.85 (0.46 to 1.58) | 0.614 |
| Aware of the main story and overdiagnosis follow-up | 63 | 42 (66.7) | 21 (33.3) | 1.05 (0.55 to 2.01) | 0.878 |
| Aware of the main story and all-cause mortality follow-up | 36 | 25 (69.4) | 11 (30.6) | 1.10 (0.49 to 2.49) | 0.819 |
| Aware of the main story only | 270 | 153 (56.7) | 117 (43.3) | 1.49 (0.98 to 2.25) | 0.062 |
| vs Unaware of the story | 243 | 156 (64.2) | 87 (35.8) | | |

*Results are adjusted for covariates: recruitment wave, age, ethnicity, marital status, highest level of education, social grade, employment status, area type, personal diagnosis of cancer, personal experience of breast screening, belief that screening is almost always a good idea, awareness of volcano news, awareness of election news, general level of trust in participants' general practitioners (GP), general level of trust in the National Health Service (NHS), breast screening intentions for next invitation. Full results of the model are reported in the online supplementary appendix 2.

negative consequences in terms of additional breast cancer deaths.

Population awareness of the breast screening news story was generally high. Television and radio were the main sources of information, broadly consistent with patterns of how most news is accessed, although the internet was used less often than observed in previous surveys.[29] Although no associations were found here, this finding is useful since it provides an estimate of the proportion of people

who may be influenced by media coverage that does have positive or negative effects on health behaviour.[7–10 12 17–19] In the absence of this study, a plausible rationale could have been found for why this estimate would be higher or lower than was shown to be the case.

Awareness of this story was related to awareness of other news stories, suggesting that an appreciable proportion of the population can be broadly dichotomised into those who are generally 'news aware' and 'news unaware'.

**Table 5** Testing for an association between awareness of the breast screening media coverage and breast screening intentions*

| Characteristic | Total (n=700) | Yes, definitely versus yes, probably; no, probably not; no, definitely not: n (%) | | Definite intention (vs no definite intention) | |
| | | No definite intention (n=99; 14.1%) | Definite intention (n=601; 85.9%) | Adjusted OR 95% CI | P value |
|---|---|---|---|---|---|
| Screening story awareness | | | | Overall: 0.108 | |
| Aware of the main story and both follow-up commentaries | 88 | 10 (11.4) | 78 (88.6) | 2.01 (0.74 to 5.48) | 0.172 |
| Aware of the main story and overdiagnosis follow-up | 63 | 4 (4.3) | 59 (93.7) | 2.66 (0.79 to 8.89) | 0.113 |
| Aware of the main story and all-cause mortality follow-up | 36 | 6 (16.7) | 30 (83.3) | 0.66 (0.20 to 2.13) | 0.486 |
| Aware of the main story only | 270 | 22 (8.1) | 248 (91.9) | 1.88 (0.99 to 3.57) | 0.054 |
| vs Unaware of the story | 243 | 57 (23.5) | 186 (76.5) | | |

*Results are adjusted for covariates: recruitment wave, age group, ethnicity, marital status, highest level of education, social grade, employment status, area type, personal diagnosis of cancer, personal experience of breast screening, belief that screening is always a good idea, awareness of volcano news, awareness of election news, general level of trust in participants' general practitioners (GP), general level of trust in the National Health Service (NHS), frequency of worry about breast cancer. Full results of the model are reported in the online supplementary appendix 2.

These results do not suggest that a notable proportion of the public are aware of health news, specifically. In contrast to these findings, recall of the main statistics was markedly low and correct responses may be largely attributable to random guessing. (Participants were asked additional questions on the extent to which they trusted the statistics and their reasons for not trusting them (if applicable). However, since responses were highly suggestive of random guessing, no further analyses of these measures were attempted.) In some respects, this is surprising since the statistics were an integral part of the story and often part of headlines (eg, refs 2–4 30) and may be a cause for concern: the number of women affected and estimated to have died as a result are important pieces of information in order for an individual to make a personal assessment of the scale and severity of the news. This finding may suggest that people either tend not to attend to or memorise this statistical information (meaning that they would not be able to factor it into their appraisal of the significance of the story) or they retain only the 'gist' of the statistics involved.[31] Awareness of the breast screening story was greater among those with higher levels of education and social grade, those who were white British and those who were older. Awareness of the breast screening news story was also lower among participants with positive attitudes towards screening (who may have been less likely to attend to a negative story).

This study has limitations. Despite the large sample size and adjustment for a range of potentially confounding variables, the number of cases was relatively small in some cells (eg, for having been invited to, but never participated in, screening and not believing, or being unsure whether screening was almost always a good idea; table 1) and some ORs were estimated with wide CIs. Real associations may not have been detected (type II error). In addition, our measures did not include a question on trust in the Breast Screening Programme, specifically, meaning that we could not test for associations with this outcome. Findings on screening uptake also relate only to anticipated future behaviour; future research could build on this study by assessing whether the announcement was followed by a decrease (or increase) in actual screening uptake. Although the response rate to this survey was higher than others of its type (eg, 71% in the present study vs 42% reported by Low et al),[32] members of the public were also less likely to participate in the survey module based on a range of characteristics for which data were available. Results may be biased, insofar as responses differed based on these variables or unmeasured participant characteristics that may have reduced population representativeness of the sample.

## CONCLUSIONS

This study found that news of errors in the Breast Screening Programme in England had reached a large proportion of the general public and that those aware of the media coverage tended to be those aware of news stories in general. The proportion of people aware was also higher among those who had more education, were in a higher social grade, or were older. In contrast, awareness of key statistics from the story was very low among participants aware of the story, even less than 6 weeks after the onset of the main media coverage. The results of this study did not provide evidence that media coverage had any effects on trust in aspects of the health service among the general public, or worry about breast cancer or breast screening intentions among women. Future research should investigate possible effects of media coverage using objective measures of screening behaviour.

**Contributors** AG, CvW and JW conceived and designed the study. AG analysed the data. AG, CvW and JW participated in the interpretation of results. AG, CvW and JW drafted the manuscript, participated in critical revision and approved the final version.

**Funding** This work was supported by a programme grant from Cancer Research UK awarded to Professor Jane Wardle (C1418/A14134). JW is supported by a Career Development Fellowship from Cancer Research UK (C7492/A17219).

**Disclaimer** Cancer Research UK was not involved in the design of this study; the collection, analysis, or interpretation of the results; in the writing of the manuscript; or in the decision to submit for publication.

**Competing interests** None declared.

**Ethics approval** Institutional ethical approval was obtained (registration number: 2951/006).

**Provenance and peer review** Not commissioned; externally peer reviewed.

**Data availability statement** No additional data are available.

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
