## [Reviewer comments · BMJ Open]

ARTICLE DETAILS

TITLE (PROVISIONAL)	Public awareness of and responses to media coverage of invitation errors in the Breast Screening Programme in England: A cross-sectional population survey
AUTHORS	Ghanouni, Alex; von Wagner, Christian; Waller, Jo

VERSION 1 - REVIEW

REVIEWER	Kirsten McCaffery University of Sydney, Australia No competing interests but I have previously published with two of the authors of this manuscript.
REVIEW RETURNED	02-Jan-2019

GENERAL COMMENTS	This study reports a national survey of population attitudes to breast screening and trust in the NHS following a news story reporting an error made by the NHS Breast Screening programme in the UK which involved the failure to issue invitations to women eligible for screening over a specified period. While I sympathize with the authors motivations for the study and understanding the relationship between health news reporting and health behaviour is important, I believe there are some problems with this study. Firstly, more explanation and justification why the study is important, its significance and clinical implications are required. These are currently missing and one is left feeling a bit 'so what'. The introduction fails to mention anything about the follow on news stories that accompanied the main story that related to all-cause mortality and ODx of breast cancer. These come as a surprise in the results and are important context for the study. Methods On what was the trust measure based – was it a validated measure, did it derive from other studies? Why were women outside the eligible screening age asked if they would be interested in breast screening? Is this really an appropriate question for women age 16 to 40 years? I am not sure it is meaningful. How do you answer this question as a 16 year old? Why was trust in the NHS generally and the GP assessed rather than or not as well as trust in the breast screening programme? This could account for why no effect was observed. Is it really plausible that trust in the entire NHS and in your GP the latter which is
---

	essentially independent of the breast screening program likely to be affected by a negative news story of the breast screening programme. I think it was a mistake not to measure this. Results Why did the analysis of quantitative estimates of the number of women affected include those people who reported not being aware of the news story? Surely their answers can only represent guesses. It would be more interesting to know what proportion of those who were aware of the story recalled the statistical information. What is usual rate of non completion of a question module in this type of survey? 30% seems a high rate of drop out and some comparison data is needed to interpret this finding. Could the analysis of awareness of the news story also be done assuming non responders were not aware of the news story to give a more accurate understanding of the penetration of the story? Why was the sample not representative by education or any other socio-economic factor other than working status. Overall I found this study a bit underwhelming and unsurprising and I am not sure that it makes much of a contribution to the literature in its current form. I think with some changes it could be improved but I still believe the authors need to do work a little more on explaining to readers why this study is important and meaningful.
--	--

REVIEWER	Philippe Autier International Prevention Research Institute
REVIEW RETURNED	03-Apr-2019

GENERAL COMMENTS	The paper of A Ghanouni and colleagues reports on a survey done after media coverage of a malfunction of the NHS computer system which led to not inviting a large number of women to breast screening (BS). The survey and its analysis did not find clear-cut profile of subjects aware or unaware of the information. However, the authors rightly comment that a type II error cannot be ruled out. Indeed, a main problem with the survey is the low number of women invited to BS who but who did not attend, ie. 56/497 (11%) when non-attendance is around 30%. Hence, the survey was depleted in women who, for a reason or another, did not attend BS, and who could have a contrasting profile and remembrance of the NHS issue. The Discussion needs to elaborate on the low proportions of non-attenders and consequences on results, including type II error. The main finding from Table 1 (awareness and belief) is puzzling. The crude odds ratio for belief that screening is a good idea is 0.78 (0.53 to 1.14). The adjusted OR is down to 0.59 (0.38 to 0.94). Why such a change in OR, ie, which variables exert such an influence on both awareness and belief? Relationships between variables on the left of Table 1 would be precious for interpretation. It could be that the marked change in OR would be due to few subjects in some cells, which entails great instability in risk estimates and associated confidence interval. If so, then the likelihood of a type II error would be high. Other comments are:
---

	1/ Tables: variables included in logistic models need to be mentioned in foot-notes. 2/ Table 4: five categories are too much in view of small numbers in entries. It would be better to do Never/occasionally vs sometimes/often/very often. 3/ I did not see ethics approval (but probably not required for a survey in the UK).
--	---

VERSION 1 – AUTHOR RESPONSE

REVIEWERS' COMMENTS TO THE AUTHORS:

REVIEWER 1

Reviewer Name: Kirsten McCaffery

Institution and Country: University of Sydney, Australia

Please state any competing interests or state 'None declared': No competing interests but I have previously published with two of the authors of this manuscript.

This study reports a national survey of population attitudes to breast screening and trust in the NHS following a news story reporting an error made by the NHS Breast Screening programme in the UK which involved the failure to issue invitations to women eligible for screening over a specified period. While I sympathize with the authors' motivations for the study and understanding the relationship between health news reporting and health behaviour is important, I believe there are some problems with this study.

Firstly, more explanation and justification why the study is important, its significance and clinical implications are required. These are currently missing and one is left feeling a bit 'so what'.

AR: We agree that this was an important omission and have made various revisions to the Introduction and Discussion with the aim of addressing this. For example, we have made the point in the Introduction that a) the prevalence of awareness is indicative of the proportion of people who may be influenced by health-related media coverage (an effect supported by other studies, if not this one) and b) prior to this study, there was uncertainty around levels of awareness. In the absence of research, it would be possible to generate equally plausible hypotheses as to why awareness was either higher or lower than observed. We have referred back to these points in the Discussion. We have also aimed to highlight the clinical significance of the study in terms of how it offers some reassurance that the media coverage did not negatively impact e.g. screening uptake (notwithstanding the caveats we refer to around possible Type II error).

The introduction fails to mention anything about the follow on news stories that accompanied the main story that related to all-cause mortality and ODX of breast cancer. These come as a surprise in the results and are important context for the study.

AR: We agree that these warrant mentioning in the Introduction as well as the Method and Results to indicate clearly that media coverage extended beyond merely reporting the Health Secretary's statement. We have added text to the Introduction stating this.

Methods: On what was the trust measure based – was it a validated measure, did it derive from other studies?

AR: We state in the Method: Trust in health services that these items were based on measures used by the Health Information National Trends Survey,²⁰ which is a routinely-administered survey in the United States that collects data from a nationally representative sample of the public. The original question wording was, "In general, how much would you trust information about cancer from each of the following?" followed by a list of items (e.g. "a doctor", "government health agencies"). Standard response options are "not at all", "a little", "some", and "a lot". We have added a reference summarising how the measure was validated.²¹

Methods: Why were women outside the eligible screening age asked if they would be interested in breast screening? Is this really an appropriate question for women age 16 to 40 years? I am not sure it is meaningful. How do you answer this question as a 16 year old?

AR: We acknowledge the issue raised by the reviewer: The younger a female participant, the further in the future their first invitation to breast screening will be and consequently the less salient the prospect of breast screening and related issues will be to them, all else being equal. This question was included and asked for all women since the current design of the Breast Screening Programme in England means that the large majority of women can expect to receive an invitation eventually and although the topic is at its least relevant to the youngest participants, we would not take it for granted that it is not relevant at all. For example, younger women may begin developing attitudes towards breast screening later in life after engaging with other screening programmes (e.g. for cervical cancer at 25 years) and also as a result of experiences of other women they know (e.g. older female family members). We carried out an exploratory analysis and found that although a lower proportion of 16-25 year old women "definitely" intended to have breast screening than women aged 41 to 95 years (69.8% vs. 91.2%), the proportions of women who "probably" and "definitely" intended to participate were very similar between age groups (93.8% vs. 95.8%), suggesting that although attitudes may differ between age groups, the strong skew towards positive attitudes is present among all ages.

Methods: Why was trust in the NHS generally and the GP assessed rather than or not as well as trust in the breast screening programme? This could account for why no effect was observed. Is it really plausible that trust in the entire NHS and in your GP the latter which is essentially independent of the breast screening program likely to be affected by a negative news story of the breast screening programme? I think it was a mistake not to measure this.

AR: We agree that the more likely effects on trust may have been with respect to the breast screening programme itself. The omission of this question was due to the way in which the original items from HINTS were adapted for the present study. HINTS measures trust in information from “a doctor” (treated as analogous to a general practitioner) and “government health agencies” (treated as analogous to the NHS). However, there is no equivalent item that could be treated as analogous to trust in screening programmes. We included a measure of behavioural intention in relation to future breast screening uptake but could have included an additional question on trust in the breast screening programme (we might also speculate that people do not necessarily perceive the screening programme as a distinct entity within the NHS which delivers it). We have noted this as a limitation in the Discussion.

Results: Why did the analysis of quantitative estimates of the number of women affected include those people who reported not being aware of the news story? Surely their answers can only represent guesses. It would be more interesting to know what proportion of those who were aware of the story recalled the statistical information.

AR: The denominator for these results is actually limited to only participants who reported being aware of the news story, for exactly the reason the reviewer mentions. We refer the reviewer to the first sentence of the “Results: Awareness of statistics from the breast screening media coverage and variables associated with awareness” section where we state “Only 233 (18.4%) of the 1,264 participants who reported being aware of the story correctly recognised the number of women who had not been invited and only 268 (21.2%) correctly recognised the estimated number of women who had their lives shortened.” We have retitled this section with the aim of making this clearer.

Results: What is usual rate of non-completion of a question module in this type of survey? 30% seems a high rate of drop out and some comparison data is needed to interpret this finding.

AR: To our knowledge, the most comparable study is that of Low et al.²⁷, which consisted of a survey based on a very similar methodology: Members of the public were approached by the same market research company to complete a set of questions on experience of symptoms of gynaecological cancer as a module within a larger routine omnibus survey. Participants were offered the option of opting out of this specific set of questions and the response rate was somewhat lower than in the present survey (42% vs. 70.6%). We have noted this in the Discussion.

Results: Could the analysis of awareness of the news story also be done assuming non-responders were not aware of the news story to give a more accurate understanding of the penetration of the story?

AR: This is an interesting idea that could be done, in principle. However, we would not be confident in any particular assumption about the awareness of non-responders. For example, we note that although white British participants were both more likely to respond to the survey and to be aware of the news story (suggesting that the sample may have an overrepresentation of aware participants),

younger participants were more likely to respond to the survey while being less likely to be aware of the story (which may mean results are skewed towards those less aware). In addition, we might speculate that the introductory text, which referred to “breast cancer screening” and “recent news stories” may have indicated the topic of the survey to those aware of the news story and who did not want to participate, resulting in them opting out.

Results: Why was the sample not representative by education or any other socio-economic factor other than working status?

AR: Representativeness of the sample was within the control of the recruiting survey company, which uses only information in the Census and Postcode Address File in order to target and approach participants. We refer the reviewer to the Methods: Design section which notes that representativeness is set with quotas based on working status, children in the household, gender, and age (i.e. more factors than working status alone). Otherwise, “classificatory variables” (e.g. social class grade) are measured as standard and additional questions (e.g. education) can be added to the survey to characterise the sample but these cannot be used to assess and control representativeness within the established methodology of the survey company. We have noted this as a limitation in the Discussion.

Overall: I found this study a bit underwhelming and unsurprising and I am not sure that it makes much of a contribution to the literature in its current form. I think with some changes it could be improved but I still believe the authors need to do work a little more on explaining to readers why this study is important and meaningful.

AR: Per our comments above, we have aimed to clarify what makes the study important, meaningful, and significant, and the clinical implications of the study.

REVIEWER 2

Reviewer Name: Philippe Autier

Institution and Country: International Prevention Research Institute

Please state any competing interests or state ‘None declared’: None declared

The paper of A Ghanouni and colleagues reports on a survey done after media coverage of a malfunction of the NHS computer system which led to not inviting a large number of women to breast screening (BS). The survey and its analysis did not find clear-cut profile of subjects aware or unaware of the information. However, the authors rightly comment that a type II error cannot be ruled out. Indeed, a main problem with the survey is the low number of women invited to BS but who did not attend, i.e. 56/497 (11%) when non-attendance is around 30%. Hence, the survey was depleted in women who, for a reason or another, did not attend BS, and who could have a contrasting profile and

remembrance of the NHS issue. The Discussion needs to elaborate on the low proportions of non-attenders and consequences on results, including type II error.

AR: We thank the reviewer for raising this interesting point. However, we would like to highlight that we measured whether invited women had ever participated in breast screening. As far as we are aware, the statistic of 30% relates to uptake of women's most recent invitation (e.g. <https://digital.nhs.uk/news-and-events/latest-news/uptake-for-routine-breast-screening-falls>). "Uptake of any invitation" is almost inevitably higher than "uptake of most recent invitation". We would also reiterate that the number of women eligible for breast screening is relatively small within the sample (26%; 497/1,894) and even if overrepresentation affected as many as 19% (30%-11%) of eligible women, this would amount to only affect approximately 5% of the sample overall. Hence, although we agree that it is possible that there is a degree of overrepresentation of participants who had undergone breast screening, we expect that this had only limited potential to bias results.

The main finding from Table 1 (awareness and belief) is puzzling. The crude odds ratio for belief that screening is a good idea is 0.78 (0.53 to 1.14). The adjusted OR is down to 0.59 (0.38 to 0.94). Why such a change in OR, i.e., which variables exert such an influence on both awareness and belief? Relationships between variables on the left of Table 1 would be precious for interpretation. It could be that the marked change in OR would be due to few subjects in some cells, which entails great instability in risk estimates and associated confidence interval. If so, then the likelihood of a type II error would be high.

AR: We have examined these associations in more depth following advice from a chartered statistician: We ran a series of analyses consisting of logistic regression models including the dependent variable (awareness of the breast screening news story) and independent variable (belief that screening was almost always a good idea), in which each of the remaining independent variables were added separately (e.g. recruitment wave and belief; age and belief). This found that the inclusion of three variables (ethnicity, awareness of volcano news, and awareness of local election news) individually increased the strength of the odds ratio of belief on awareness to a relatively large degree. Hence, collectively (i.e. based on a model with independent variables of ethnicity, awareness of volcano news, awareness of local election news, and belief), the odds ratio of the belief variable was strengthened to 0.60, very similar to that of the final, fully-adjusted model. This is due to these three variables having a positive association with awareness (in the opposite direction to that of beliefs) and hence having a synergistic effect on this relationship in the fully-adjusted logistic regression model.

Other comments are:

1/ Tables: variables included in logistic models need to be mentioned in foot-notes.

AR: We have added a footnote to Tables 3-5, listing the relevant variables.

2/ Table 4: five categories are too much in view of small numbers in entries. It would be better to do Never/occasionally vs sometimes/often/very often.

AR: We agree with the reviewer's point. We have updated the Results accordingly, which do not change the conclusions of the study.

3/ I did not see ethics approval (but probably not required for a survey in the UK).

AR: We refer the reviewer to the first sentence of the Methods: Design section, in which we state that "Institutional ethical approval was obtained (registration number: 2951/006)."

VERSION 2 – REVIEW

REVIEWER	Kirsten McCaffery University of Sydney
REVIEW RETURNED	09-Jun-2019

GENERAL COMMENTS	This version is much improved. Just a few very minor suggestions. Change 36.0% in the abstract to 36%. The limitations should state more clearly that we cannot say if trust in breast screening is undermined by the news story as it was not assessed. In the introduction please give a little more detail about the coverage of the flexible sigmoidoscopy trial which increased uptake of bowel screening.
---

REVIEWER	Philippe Autier International Prevention Research Institute
REVIEW RETURNED	28-May-2019

GENERAL COMMENTS	The revised paper of A Ganouni and colleagues has somewhat improved. However, I have the impression that authors experience difficulties in understanding how low numbers of subjects in some table cells can dramatically affect results and lead to type II error. Central to this issue is the very low number (n=55) of women in eligible age groups who never attended screening, who are underrepresented. A similar remark is applicable to the belief question (n=143). The percentages are not important. What matters are the absolute numbers as these represent the data feeding statistical analyses. The instability of odds ratios after inclusion of some variables in logistic models is therefore unsurprising and denotes a high probability of type II errors. The Discussion section does not clearly mention the possibility of type II errors. Authors should recall what type II error is and explain how it can affect study results and their interpretation.
---

VERSION 2 – AUTHOR RESPONSE

REVIEWERS' COMMENTS TO THE AUTHORS:

REVIEWER 1

Reviewer Name: Kirsten McCaffery

Institution and Country: University of Sydney, Australia

Please state any competing interests or state 'None declared': No competing interests but I have previously published with two of the authors of this manuscript.

This version is much improved. Just a few very minor suggestions. Change 36.0% in the abstract to 36%. The limitations should state more clearly that we cannot say if trust in breast screening is undermined by the news story as it was not assessed. In the introduction please give a little more detail about the coverage of the flexible sigmoidoscopy trial which increased uptake of bowel screening.

Authors' Response: We are pleased to hear the reviewer believes the draft has much improved now, and we thank them for their helpful suggestions. We have changed "36.0%" in the abstract to "36%" (and "66.7%" to "67%" for consistency). We would like to highlight the text in the Discussion which states that "our measures did not include a question on trust in the Breast Screening Programme, specifically, meaning that we could not test for associations with this outcome", which we believe addresses the reviewer's second comment. We have also provided information on what was included in the coverage of the flexible sigmoidoscopy trial with references.

REVIEWER 2

Reviewer Name: Philippe Autier

Institution and Country: International Prevention Research Institute

Please state any competing interests or state 'None declared': None declared

The revised paper of A Ganouni and colleagues has somewhat improved. However, I have the impression that authors experience difficulties in understanding how low numbers of subjects in some table cells can dramatically affect results and lead to type II error. Central to this issue is the very low number (n=55) of women in eligible age groups who never attended screening, who are underrepresented. A similar remark is applicable to the belief question (n=143). The percentages are not important. What matters are the absolute numbers as these represent the data feeding statistical analyses. The instability of odds ratios after inclusion of some variables in logistic models is therefore unsurprising and denotes a high probability of type II errors. The Discussion section does not clearly mention the possibility of type II errors. Authors should recall what type II error is and explain how it can affect study results and their interpretation.

AR: We have rephrased the acknowledgement of the possibility of Type II error in the limitations section of the Discussion, where we state that “the number of cases was relatively small in some cells (e.g. for having been invited to, but never participated in, screening and not believing, or being unsure whether, screening was almost always a good idea; Table 1) and some odds ratios were estimated with wide confidence intervals. Real associations may not have been detected (Type II error)”. We also refer to the possibility of Type II error in the table of strengths and limitations of the study.